# Autonomous Controller-Aware Scheduling of Intra-Platoon V2V Communications

**DOI:** 10.3390/s23010060

**Published:** 2022-12-21

**Authors:** Paweł Sroka, Erik Ström, Tommy Svensson, Adrian Kliks

**Affiliations:** 1Communication Systems Group, Department of Electrical Engineering, Chalmers University of Technology, 412 96 Gothenburg, Sweden; 2Institute of Radiocommunications, Poznan University of Technology, 60-965 Poznan, Poland

**Keywords:** platooning, autonomous vehicles, CACC, scheduling

## Abstract

In this paper, we investigate the problem of reducing the use of radio resources for vehicle-to-vehicle communications in an autonomous platooning scenario. Achieving reliable communications, which is a key element allowing for the tight coordination of platoon vehicles’ motion, might be challenging in a case of heavy road traffic. Thus, in this paper, we propose to reduce the number of intra-platoon transmissions required to facilitate the safe autonomous control of vehicle mobility, by analyzing the impact of cars’ behaviors (in terms of acceleration changes) on the evolution of the inter-vehicle distance errors within the platoon. We derive formulas representing the relation between the platoon leader’s acceleration changes and the evolution of the distance error, velocity difference, and the accelerations for the first pair of vehicles. Furthermore, we propose a heuristic algorithm for selection of the intra-platoon messaging period for each platoon vehicle that minimizes the use of radio resources subject to the safety constraint, represented as the fraction of the total time when emergency braking is activated. The presented simulation results indicate that the proposed approach is capable of ensuring safe platoon operation and simultaneously providing a significant reduction in the use of resources, compared with conventional fixed-period transmission.

## 1. Introduction

One of the main concerns related to the rapid development of advanced contemporary cars is to increase the safety and the efficiency of road traffic, while maintaining the comfort level of the passengers. Various solutions have been proposed towards the realization of these paradigms, just to mention the Anti-lock Braking Systems (ABS), Electronic Stability Program (ESP), Anti-Slip Regulation (ASR), but also lane assist systems, pre-collision assist, or the Adaptive Cruise Control (ACC) system. With so many functions introduced, cars become more and more automated, where many aspects are now controlled by the on-board computer. This brings us closer to the realization of the idea of autonomous cars, where the full car control will be realized by the on-board systems.

One of the vital use cases for autonomous vehicles, aiming at increasing road efficiency, is vehicles platooning, where a group of vehicles forms a convoy led by the platoon leader. The use of a platoon provides various gains, including an increase in road capacity [1], a reduction in fuel consumption [2], and, consequently, lower CO${}_{2}$ emissions [3]. However, in order to benefit from platooning, the inter-vehicle distances in a convoy need to be relatively small, which requires the use of a precise autonomous controller. Several solutions aiming at automatic vehicle control in platooning scenario have been proposed, such as the Cooperative Adaptive Cruise Control (CACC) [4,5], or its modifications called platoon controllers [6,7]. With such a controller, each vehicle is supposed to follow the pre-defined behavior (e.g., maintaining the targeted inter-vehicle spacing) by controlling its speed and acceleration. In any case, the CACC or the platoon controller requires the use of wireless communications to exchange information between vehicles, as the data retrieved with on-board sensors are insufficient to maintain the required level of safety. It has been shown that reliable communications within the platoon can lead to a significant reduction in inter-car distances [8].

While Vehicle-to-Vehicle (V2V) and Vehicle-to-Everything (V2X) communication is a key enabling technology in platooning, it is tightly coupled with vehicle dynamics control. Any errors in the exchange of information in the platoon may significantly affect the platooning operations, thus jeopardising the road safety [9]. The platoon controller should achieve string stability, ensuring the attenuation of the spacing errors between vehicles as they propagate downstream along the platoon. Furthermore, to maintain the short inter-vehicle gaps, a short response time of the control system is required, which is challenging, as continuous-time provisioning of vehicular data to the platoon controller is practically not possible. Typically, sampled-data control is used, where the data packets containing the needed information are available only at discrete time instants. Thus, to enable the short response time of the control system, the frequent periodic dissemination of information using wireless communication is typically assumed [10]. However, the existing periodic communication mechanisms usually do not account for the constrained amount of wireless communication resources when designing and evaluating the controller. It has been shown in [9] that, due to the limited availability of resources, both with the Dedicated Short-Range Communications (DSRC) and the Cellular-V2X (C-V2X) packet drops may occur in unfavorable scenarios with heavy road traffic. Thus, the efficient use of resources, allowing for the maximization of the number of communicating vehicles, is of high importance. With the efficient scheduling of data transmissions between the intercommunicating platoon vehicles, the scarce communication resources can be used economically. However, such efficient and reliable scheduling, and dissemination of data is a very challenging task, as reducing the frequency of intra-platoon transmissions might result in degradation of the platoon control performance.

Accounting for the problem of constrained wireless resources, in this paper, we investigate the idea of adaptive adjustment of the rate of intra-platoon transmissions while maintaining the required safety level of autonomous platooning. Our aim is to reduce the use of wireless resources with the constraint on the minimum distance kept between the platoon vehicles, while using a fixed controller design proposed in [6]. We extend the idea of event-triggered scheduling introduced in [11], by planning the transmission of each platoon vehicle, taking into account the predicted evolution of the inter-vehicle distances depending on the introduced latency of the disseminated information. We formulate the optimization problem and propose a heuristic solution based on exhaustive search that allows for the selection of the transmission intervals while keeping the inter-vehicle distances above the predefined threshold. We evaluate this proposal in simulations implemented with MATLAB, comparing the consumption of resources and the ability to fulfill the minimum distance constraint with a conventional transmission scheme that uses a fixed-period transmission. We show that it is possible to maintain a safe platoon operation while reducing the use of wireless resources compared to a fixed-period approach, by taking into account the design and parameters of the autonomous controller used in platoon vehicles.

The main contributions of this paper can be summarized as follows:We formulate the optimization problem, focusing on the minimization of the use of resources, with the constraint on the fraction of time where violations of the minimum inter-vehicle distance were experienced.We derive a recursive formula for the calculation of the distance error and the relative velocity between the platoon leader and the first follower, as well as for the acceleration of the first follower calculated according to the control strategy proposed in [6].We propose a heuristic solution to find the maximum period and the initial delay of transmission for each platoon vehicle, accounting for the constraint on the minimum distance. We use prediction of evolution of the distance error between two subsequent platoon vehicles over time, taking into account the availability of a predefined set of possible transmission periods and initial delays.We evaluate the proposed approach in simulations, showing that it is possible to reduce the use of resources compared to a fixed-period transmission scheme.

The paper is structured as follows. In Section 2, a short overview of the existing solutions, focusing on platooning control, with communication imperfections or constraints, is given. In Section 3, we present the considered system model and the considered platoon controller. Section 4 introduces the formulation of the optimization problem with the constraint on the minimum inter-vehicle distance violations. This section also presents a recursive formula for the calculation of the evolution of the distance error and the relative velocity between the platoon leader and the first follower, based on the changes of leader acceleration. It is followed by a presentation of the proposed heuristic solution in Section 5, employing the prediction of the distance error evolution over time when selecting the transmission period and initial delay. Subsequently, the results of numerical simulation are presented in Section 6, comparing the proposed heuristic approach with a conventional fixed-period transmission. Finally, the discussion of the observations concludes this work in Section 7.

## 2. Related Work

The area of autonomous platooning supported by wireless communications has gained a significant focus in the literature in recent years. Many works have been published that can be mostly categorized into one of the following two areas:Design and performance analysis of the inter-vehicle communication network, including the study on the efficient exchange of information within the platoon [12], interference mitigation [13,14], or the transmission delay analysis [15].Design and stability analysis of platoon control strategies [5,6,16,17].

However, the main limitation of these works is that they focus only on one area of platoon control, neglecting the impact of varying performance of the other one. The communication-centric works typically abstract the control system (assuming that platoon stability can be maintained), while the control-centric works assume a deterministic performance and behavior from the communication network.

Several works can be found that consider the impact of wireless communications performance on the control system design. Such an approach is considered in [18], accounting for the latency in the distribution of information between vehicles, which is derived using queuing theory. Similarly, in [19], the authors propose a graceful degradation mechanism for platooning control, switching from CACC to ACC when a failure in wireless communications is experienced. The fallback mechanism for switching between CACC and ACC is also considered in [20]. The main disadvantage of these works is the limited model of wireless communications, accounting only for the delay in acquiring information or random packet drops.

A more communication-oriented co-design approach can be found in [21], where the transmission scheduling based on most regular binary sequences and priority-based platoon control is proposed. However, the proposed transmission scheduling assumes static allocation once the proper patterns are found, thus not accounting for the random nature of the channel access that is typically experienced in wireless communications. A scheduling design for C-V2X communications in platoons is proposed in [22], with the aim being to minimize the tracking error of the inter-vehicle distance, velocity, and acceleration. Along with the scheduling mechanism, a platoon controller coefficients adaptation is considered to fulfill the optimization objective. Similarly, a co-design based on multi-hop communications and distributed controller parameters adaptation is proposed in [23]. A different control design is considered in [24], where an age-of-information-based communications scheduling is proposed for an infrastructure-based networked estimator and platoon controller.

While the solutions mentioned above aim to consider jointly the platoon control and the communication model, typically, they consider a very simplified mechanism of services providing data packets to be transmitted in regular intervals. As such a periodic model is typically assumed e.g., for the Cooperative Awareness Messages (CAMs) in V2X communications, it might not be a correct one for the platooning scenario. Typically, the platoon controller will require exchanges of information on a quasi-periodic basis, depending on the occurrence of certain events (e.g., sudden deceleration, detecting obstacles, etc.). Such an approach is considered in [10,25], where a co-design of event-triggered scheduling of communications and platoon control is proposed. However, both works focus mostly on the control design, abstracting the wireless communications model.

An event- or scenario-based adaptation of control mechanisms is also considered in [26], where a multi-layer model predictive control for platooning is proposed, which makes use of the Dynamic Congestion Control (DCC) mechanism to adapt the messaging rate. However, the proposed adaptation relies only on a set of fixed configurations, which are selected based on the estimated channel load. In order to account for the changes of vehicles’ motion parameters with the platoon control, event-based message generation rules can be considered, which rely on observed changes of the ego or of surrounding vehicles. Such generation rules for the cooperative perception mechanism are discussed in [27], with the application of a similar approach in platooning scenario being presented in [28], where specific service profiles are considered for message generation.

Following the idea of an event-triggered scheduling of communications and the need to address specific features of wireless communications, such as the use of scheduling grants in C-V2X, we consider in this work an event-based design of quasi-periodic scheduling of transmissions, where the interval between the transmission of subsequent messages for a platoon vehicle is dynamically selected based on the road situation.

## 3. System Model

Let us assume a motorway scenario where a platoon of *N* vehicles (led by the so-called leader vehicle) drives in a coordinated way, with the longitudinal motion being controlled using the autonomous platoon controller, as shown in Figure 1. The system model can be then described in a discrete way, with the time step of Δt, using the following formulas.
(1)s(k+1)=As(k)+u(k)+z(k),
where:s(k) is the system state vector at time kΔt, defined as:(2)$$s\left(k\right)={\left[\begin{array}{c} x_{0}\left(k\right),v_{0}\left(k\right),a_{0}\left(k\right),x_{1}\left(k\right),v_{1}\left(k\right),a_{1}\left(k\right),\ldots ,x_{N-1}\left(k\right),v_{N-1}\left(k\right),a_{N-1}\left(k\right) \end{array}\right]}^{T},$$where $x_{i}$, $v_{i}$, and $a_{i}$ stand for the longitudinal position, velocity, and acceleration of vehicle *i*, respectively.A is the state transition matrix, representing the position, velocity. and acceleration changes according to the accelerated motion law:(3)$$A=\left[\begin{array}{cccccccccc} 1 & \Delta t & \frac{{\Delta t}^{2}}{2} & 0 & 0 & 0 & \ldots & 0 & 0 & 0\\ 0 & 1 & \Delta t & 0 & 0 & 0 & \ldots & 0 & 0 & 0\\ 0 & 0 & 1 & 0 & 0 & 0 & \ldots & 0 & 0 & 0\\ 0 & 0 & 0 & 1 & \Delta t & \frac{{\Delta t}^{2}}{2} & \ldots & 0 & 0 & 0\\ 0 & 0 & 0 & 0 & 1 & \Delta t & \ldots & 0 & 0 & 0\\ 0 & 0 & 0 & 0 & 0 & 1 & \ldots & 0 & 0 & 0\\ \vdots & \vdots & \vdots & \vdots & \vdots & \vdots & \ddots & \vdots & \vdots & \vdots \\ 0 & 0 & 0 & 0 & 0 & 0 & \ldots & 1 & \Delta t & \frac{{\Delta t}^{2}}{2}\\ 0 & 0 & 0 & 0 & 0 & 0 & \ldots & 0 & 1 & \Delta t\\ 0 & 0 & 0 & 0 & 0 & 0 & \ldots & 0 & 0 & 1 \end{array}\right],$$u(k) is the controller actions vector (acceleration changes).$z\left(k\right)={\left[z_{1}\left(k\right),z_{2}\left(k\right),\ldots ,z_{3N}\left(k\right)\right]}^{T}$ is the disturbances vector. In this work, we focus only on a case where the acceleration disturbance (acceleration change due to external reasons) may occur only to the leader vehicle, so: $\forall_{i\neq 3}\;z_{i}\left(k\right)=0$. A more general scenario with acceleration disturbances being applied also to other platoon vehicles is left for future study.

**Figure 1 sensors-23-00060-f001:**
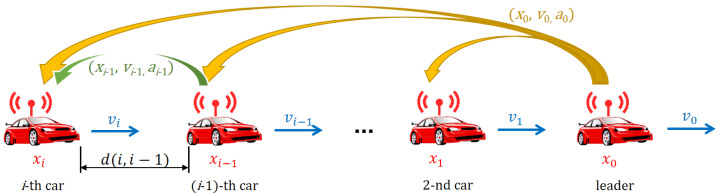
Considered platooning scenario using the autonomous leader-predecessor-follower controller and communications model.

We assume that all platoon vehicles employ a CACC platoon controller proposed in [6], where a leader-predecessor-follower model is considered. Therefore, each platoon vehicle, except for the leader, updates its acceleration based on the mobility information received from both the platoon leader and its predecessor. The controller actions u(k) are then calculated according to the formula:(4)$$u_{j}\left(k\right)=\left\{\begin{array}{cc} \max\left\{\min\left\{{\hat{a}}_{i}\left(k\right),a_{\max}\right\},a_{\min}\right\}-a_{i}\left(k\right) & \mathrm{if}\;j=3(i+1),i=0,\ldots ,N-1\\ 0 & \mathrm{otherwise} \end{array}\right.,$$
with $a_{\min}$ and $a_{\max}$ denoting the minimum and maximum acceleration constraints, respectively, and ${\hat{a}}_{i}\left(k\right)$ representing the autonomous controller-calculated acceleration:
(5)$$\begin{array}{cc} {\hat{a}}_{i}\left(k\right)= & \overset{\mathrm{Location}\;\mathrm{update}}{\overset{︷}{\alpha_{1}\left(d_{d}-x_{i-1}\left(k\right)+x_{i}\left(k\right)\right)}}-\overset{\mathrm{Velocity}\;\mathrm{update}}{\overset{︷}{\alpha_{2}\left(v_{i-1}\left(k\right)-v_{i}\left(k\right)\right)}}\\ & -\underset{\mathrm{Leader}-\mathrm{related}\;\mathrm{velocity}\;\mathrm{update}}{\underset{︸}{\alpha_{3}\left(v_{0}\left(k\right)-v_{i}\left(k\right)\right)}}+\underset{\mathrm{Acceleration}\;\mathrm{update}}{\underset{︸}{\alpha_{4}a_{i-1}\left(k\right)+\alpha_{5}a_{0}\left(k\right)-a_{i}\left(k\right)}}, \end{array}$$
where $\alpha_{l},l\in \left\{1,2,3,4,5\right\}$ are the controller coefficients (constants). Furthermore, we assume an ideal engine model, applying immediately the requested acceleration (the impacts of engine imperfections and their modeling are studied e.g., in [29]). The first term in (Equation 5), i.e., $\alpha_{1}\left(d_{d}-x_{i-1}\left(k\right)+x_{i}\left(k\right)\right)$ represents the impact of the inter-vehicle distance error. The second and third terms represent the influence of the velocity difference vs. the preceding and the leader vehicle, respectively. Similarly, the fourth and fifth terms indicate the impact of the preceding and leader vehicles’ accelerations.

Such a controller definition assumes that the updates are applied in a continuous manner. However, in a realistic platooning scenario, the information on the mobility of the platoon leader or the preceding car needs to be transmitted using wireless communication, and is updated only when a new intra-platoon message is received. Therefore, we assume that the controller actions are made in an event-based manner, whenever a new message updating the mobility parameters is received.

In this work, we consider a general communication scheme based on the quasi-periodic exchange of information, where the interval between subsequent messages may be selected dynamically based on detected events and vehicles’ motion parameters, as shown in Figure 2.

Each vehicle, upon the detection of any event resulting in a change of motion parameters, selects the initial latency τ of the first transmitted message and the interval between any following messages $\Delta t_{Tx}$. The communication pattern (resources used) for each vehicle is then specified with (Equation 6).
(6)$$q_{s}^{\left(i\right)}\left(k\right)=\left\{\begin{array}{cc} 1 & \;\mathrm{if}\;\mathrm{vehicle}\;i\;\mathrm{is}\;\mathrm{transmitting}\;\mathrm{at}\;\mathrm{time}\;k\Delta t\\ 0 & \mathrm{otherwise}. \end{array}\right.$$

The specific V2V communication interface is out of the scope of this paper; however, one can consider both the DSRC or C-V2X being applied, as both are suitable in the provisioning of such information exchange. DSRC is well suited to event-based transmission, and typically, assuming limited range communications within the platoon, it experiences small latency [30]. On the other hand, the introduced delay may vary for consecutive packets, due to the channel access method used. C-V2X is much more predictable in terms of latency, as it uses a grant mechanism, where the sub-channels available for transmission are allocated in advance for a specific period; however, it is less suitable for handling event-based messaging, thus introducing greater delays.

For the considered scenario, we assume that each vehicle is equipped with an emergency braking mechanism that becomes activated if the distance to the preceding vehicle is below a certain threshold $d_{Th}$.

## 4. Problem Formulation

Contrary to other works, rather than optimizing the controller while knowing the communications performance, we aim to minimize the amount of resources used for intra-platoon communications. In particular, we focus on the minimization of the total number of transmissions while maintaining the safe operation of the platoon using a known controller, understood as keeping the time of the emergency braking activation below a predefined level ϵ. More specifically, we assume that emergency braking is activated when the distance between two subsequent vehicles is below a specified threshold distance $d_{Th}$. Thus, the optimization problem in terms of the information update period ($\Delta t_{Tx}$) and the initial latency (τ) can be described as:(7)$$\min_{\tau ,\Delta t_{Tx}}\sum_{i=0}^{N-1}\sum_{k}q_{s}^{\left(i\right)}\left(k\right),$$
subject to:(8)$$\forall_{i=0,\ldots ,N-1}\;\frac{1}{T}\int_{0}^{T}I\left[e_{i}\left(t\right)>d_{d}-d_{Th}\right]dt<\epsilon ,$$
where I() is the indicator function, defined as:(9)$$I\left[e_{i}\left(t\right)>d_{d}-d_{Th}\right]=\left\{\begin{array}{cc} 1, & \mathrm{if}\;e_{i}\left(t\right)>d_{d}-d_{Th}\\ 0, & \mathrm{otherwise} \end{array}\right.,$$

Here, *T* is the observation interval (so with the discrete model, we sum over $k=\frac{T}{\Delta t}$ samples), ϵ is the acceptable fraction of emergency braking events, and $e_{i}\left(t\right)$ is the distance error between two subsequent platoon vehicles at time *t*, defined as $e_{i}\left(t\right)=d_{d}-\left(x_{i-1}\left(t\right)-x_{i}\left(t\right)\right)$. The key element impacting upon the selection of resources is then the safety constraint defined in (Equation 8). However, the exact solution to (Equation 8) is nontrivial, due to the mutual dependencies between the accelerations of the consecutive platoon vehicles introduced by the controller, and the possibility of different latency introduced by the individual platoon vehicles. Such a dependency model between the acceleration disturbance $z_{3}\left(k\right)$ and the distance error for the first pair of vehicles in the platoon $e_{1}\left(k\right)$ has been found, assuming an information update at every time step Δt, following the rationale presented in Appendix A, along with the formulas for the velocity difference in the first pair $\Delta v_{0,1}\left(k\right)$ and the acceleration of the first follower $a_{1}\left(k\right)$. Such relations as functions of the information update period (Δt) can be expressed as:(10)$$\begin{array}{cc} \; & e_{1}\left(k\right)=-\frac{{\Delta t}^{2}}{2}\sum_{j=0}^{k-1}\left(z_{3}\left(k-1-j\right)\cdot \sum_{m=0}^{\left\lfloor \frac{k-1-j}{2}\right\rfloor }\sum_{n=0}^{\lfloor \frac{k-1-j}{2}\rfloor -m}\theta_{m,n}^{\left(e\right)}\left(k-j\right)\cdot {\left(\left(\alpha_{2}+\alpha_{3}\right)\Delta t\right)}^{m}\cdot {\left(\alpha_{1}\frac{{\Delta t}^{2}}{2}\right)}^{n}\right),\\ \; & \Delta v_{0,1}\left(k\right)=\Delta t\sum_{j=0}^{k-1}\left(z_{3}\left(k-1-j\right)\cdot \sum_{m=0}^{\left\lfloor \frac{k-1-j}{2}\right\rfloor }\sum_{n=0}^{\lfloor \frac{k-1-j}{2}\rfloor -m}\theta_{m,n}^{\left(v\right)}\left(k-j\right)\cdot {\left(\left(\alpha_{2}+\alpha_{3}\right)\Delta t\right)}^{m}\cdot {\left(\alpha_{1}\frac{{\Delta t}^{2}}{2}\right)}^{n}\right),\\ \; & a_{1}\left(k\right)=\sum_{j=0}^{k-1}\left(z_{3}\left(k-1-j\right)\cdot \sum_{m=0}^{\left\lfloor \frac{k-j}{2}\right\rfloor }\sum_{n=0}^{\lfloor \frac{k-j}{2}\rfloor -m}\theta_{m,n}^{\left(a\right)}\left(k-j\right)\cdot {\left(\left(\alpha_{2}+\alpha_{3}\right)\Delta t\right)}^{m}\cdot {\left(\alpha_{1}\frac{{\Delta t}^{2}}{2}\right)}^{n}\right), \end{array}$$
where the corresponding $\theta_{m,n}^{\left(.\right)}\left(k\right)$ coefficients are calculated recursively as follows.
$$\begin{array}{cc} \; & \theta_{m,n}^{\left(e\right)}\left(k\right)=\theta_{m,n}^{\left(e\right)}\left(k\right)+2\sum_{i=1}^{k-1}\theta_{m,n}^{\left(v\right)}\left(i\right),\\ \; & \theta_{m,n}^{\left(v\right)}\left(k\right)=-\sum_{i=1}^{k-1}\theta_{m,n}^{\left(a\right)}\left(i\right),\\ \; & \theta_{m,n}^{\left(a\right)}\left(k\right)=-\theta_{m-1,n}^{\left(v\right)}\left(k\right)-\theta_{m,n-1}^{\left(e\right)}\left(k\right), \end{array}$$
with $\theta_{0,0}^{\left(e\right)}\left(k\right)=2k-1$, $\theta_{0,0}^{\left(v\right)}\left(k\right)=1$, $\theta_{0,0}^{\left(a\right)}\left(k\right)=1$, $\theta_{m,0}^{\left(a\right)}\left(k\right)=-\theta_{m-1,0}^{\left(v\right)}\left(k-1\right)$, $\theta_{0,n}^{\left(a\right)}\left(k\right)=-\theta_{0,n-1}^{\left(e\right)}\left(k-1\right)$.

Figure 3 and Figure 4 show the distance error calculated using the above model and simulated with the considered platoon controller for a single disturbance ($z_{3}\left(0\right)=2\;{m/s}^{2}$) and for random disturbances over time, respectively. The results clearly indicate the correctness of the derived model (i.e., both the curves for the simulation and the theoretical model overlap ideally).

The derivation can be continued in a similar way to obtain the distance error representations for other vehicle pairs. However, one should note that here, the relations will be more complicated, as these will account for the information received from different sources (the platoon leader and the preceding car). Hence, the impact of leader acceleration disturbance on the distance error will depend on the vehicle position in the platoon, as well as on the individual latencies when passing the information between the vehicles. One should also note that (Equation 10) assumes a fixed and equal delay when transmitting the intra-platoon messages, which might not be the case in a real scenario. Furthermore, the problem defined in (Equation 7) is a mixed-integer programming problem, where the constraint (Equation 8) relies on indicator function values ($I\left[e_{i}\left(t\right)>d_{d}-d_{Th}\right]$), which are discrete (in fact, Boolean). Hence, the optimal solution cannot be easily found, even when applying relaxation techniques. Therefore, in the next section, we propose a heuristic approach that is based on the iterative prediction of the distance error.

## 5. Heuristic Approach

As mentioned in the previous section, the optimal solution to (Equation 7) cannot, in general, be easily found. Thus, we propose a heuristic solution based on the constraint (Equation 8) and based on relaxing the integer problem by finding the solution {${\Delta t_{Tx}}^{sel}$, $\tau^{sel}$} from the available values in the finite sets $T_{d}$ and $T_{\tau }$, respectively, that maximizes the time period until the violation of the minimum distance $d_{Th}$. We assume that each transmitting vehicle performs such a heuristic search individually, having knowledge on the information transmitted by the leader and on the motion parameters of the following vehicle. The selection process, described in Algorithm 1, relies on the prediction of the distance error $e_{i}\left(k\right)$ evolution for a given pair of vehicles over a predefined time span $t_{Th}$. The distance between vehicles i−1 and *i* is calculated, assuming the update of information used in the acceleration controller for the selected values $\Delta t_{l}\in T_{d}$ and $\tau_{m}\in T_{\tau }$. For each prediction step $\Delta t_{l}$, the predicted position, velocity, and acceleration of the considered vehicles are updated according to the accelerated motion law, and the distance error is calculated. The prediction is performed as long as the distance between vehicles is greater than the emergency braking threshold $d_{Th}$. If at any prediction point such a distance constraint is violated, then the resulting time span value *t* is stored for the evaluated set {${\Delta t}_{l}$, $\tau_{m}$}. Otherwise, an infinite time span is assumed when at any point, the acceleration and velocity of the following vehicle exceeds the acceleration and velocity of the preceding vehicle, or when $t_{Th}$ is used if the distance remains above $d_{Th}$ for the whole prediction window. Finally, the solution is chosen as the pair maximizing the predicted time span.
**Algorithm 1** Exhaustive search algorithm1:**procedure**Find Tx period($v_{0}$, $a_{0}$, $x_{i-1}$, $v_{i-1}$, $a_{i-1}$, $x_{i}$, $v_{i}$, $a_{i}$, $d_{d}$, $d_{Th}$, $T_{d}$, $T_{\tau }$, $t_{Th}$, f(.))2:                ▹$x_{i}$—long. position of vehicle *i*, $v_{i}$—velocity of vehicle *i*, $a_{i}$—acceleration of vehicle *i*, $d_{d}$—target inter-vehicle distance, $d_{Th}$—minimum distance constraint, $T_{d}$—set of possible transmission periods, $T_{\tau }$—set of possible initial time offsets, $t_{Th}$—maximum prediction time span, f(.)—controller function3:    **for** each $\Delta t_{l}\in T_{d}$, l=1,…,|T| **do**:4:        **for** each $\tau_{m}\in T_{\tau }$, $m=1,\ldots ,|T_{\tau }|$ **do**:5:           Calculate:6:           $x_{i-1}^{*}=x_{i-1}+v_{i-1}\tau_{m}+a_{i-1}\frac{\tau_{m}^{2}}{2}$, $v_{i-1}^{*}=v_{i-1}+a_{i-1}\tau_{m}$7:           $x_{i}^{*}=x_{i}+v_{i}\tau_{m}+a_{i}\frac{\tau_{m}^{2}}{2}$, $v_{i}^{*}=v_{i}+a_{i}\tau_{m}$,8:           $v_{0}^{*}=v_{0}+a_{0}\tau_{m}$,9:           $a_{i}^{*}=f\left(v_{0}^{*},a_{0},x_{i-1}^{*},v_{i-1}^{*},a_{i-1},x_{i}^{*},v_{i}^{*},a_{i}^{*},d_{d}\right)$,10:          $d^{*}=x_{i-1}^{*}-x_{i}^{*}$, t=τ11:           **while** $d^{*}>d_{Th}$ & $t<t_{Th}$ & $v_{i}^{*}>0$ **do**:12:               Calculate:13:               $x_{i-1}^{*}=x_{i-1}^{*}+v_{i-1}^{*}{\Delta t}_{l}+a_{i-1}\frac{{\Delta t}_{l}^{2}}{2}$, $v_{i-1}^{*}=v_{i-1}^{*}+a_{i-1}{\Delta t}_{l}$14:               $x_{i}^{*}=x_{i}^{*}+v_{i}^{*}{\Delta t}_{l}+a_{i}^{*}\frac{{\Delta t}_{l}^{2}}{2}$, $v_{i}^{*}=v_{i}^{*}+a_{i}^{*}{\Delta t}_{l}$,15:               $v_{0}^{*}=v_{0}^{*}+a_{0}{\Delta t}_{l}$16:               $d^{*}=x_{i-1}^{*}-x_{i}^{*}$, $t=t+{\Delta t}_{l}$,17:               $a_{i}^{*}=f\left(v_{0}^{*},a_{0},x_{i-1}^{*},v_{i-1}^{*},a_{i-1},x_{i}^{*},v_{i}^{*},a_{i}^{*},d_{d}\right)$18:               **if** $a_{i-1}^{*}-a_{i}^{*}>0$ & $v_{i-1}^{*}-v_{i}^{*}>0$ & $d^{*}>d_{Th}$ **then**:19:                   t=∞20:                   **break**21:               **end if**22:           **end while**23:           Set: $t_{m}^{*}=t$24:        **end for**25:        Select: $m^{*}=\arg\max_{m}t_{m}^{*}$26:        Set: $\tau_{l}^{*}=\tau_{m^{*}}$27:        Set: $t_{l}^{**}=t_{m^{*}}^{*}$28:    **end for**29:    Select: $l^{*}=\arg\max_{l}t_{l}^{**}$30:    Set: $\tau^{\left(sel\right)}=\tau_{l^{*}}^{*}$31:    Set: ${\Delta t_{Tx}}^{\left(sel\right)}={\Delta t}_{l^{*}}$**return**$\left(\tau^{\left(sel\right)},{\Delta t_{Tx}}^{\left(sel\right)}\right)$32:**end procedure**

The approach proposed in Algorithm 1 provides a suboptimal solution, as it operates only with finite sets of $T_{d}$ and $T_{\tau }$. Furthermore, it assumes accurate knowledge of the current motion parameters of the preceding and the following vehicle, as well as an error-free exchange of information. Therefore, some emergency braking distance violations can be experienced even when it is in use, as the algorithm aims only at minimizing a chance of such an event. The number of such occurrences will depend on the acceleration disturbance changes and the sets of available values $T_{d}$ and $T_{\tau }$.

The approach proposed with Algorithm 1 can be further extended by applying a hysteresis-based approach, where the final values of the messaging period are selected as the minimal value out of the past values, assuming a certain memory depth of *r* milliseconds. The aim of the use of hysteresis is to increase the robustness of the proposed solution to sudden and frequent acceleration changes that may lead to a loss of stability when less frequent messaging is used.

## 6. Numerical Results

The proposed heuristic approach has been evaluated in numerical simulations using MATLAB, as it allows us to abstract the communication layer and separate it from the platoon control model. These were performed according to the model described in Section 3. We assumed a platoon of six vehicles moving on a straight road section with an initial velocity of 20 m/s and a desired inter-vehicle spacing $d_{d}$ of 3 m, with the leader changing its acceleration according to the disturbance process z(t). The acceleration disturbance z(t) has been modeled as a random process, where the disturbance is applied with an inter-arrival time being an exponentially distributed random variable, and the amplitude of z(t) being a uniformly distributed random variable in an interval $\left[z_{min},z_{max}\right]$. Figure 5 shows an example of the changes of a leader’s acceleration over a single simulation run duration.

Each platoon vehicle broadcasted its motion parameters (position, velocity, and acceleration) periodically, with the interval being selected according to one of the following two transmission policies:Fixed periodic transmission—each vehicle broadcasts messages with the same pre-selected interval for the whole simulation duration.Adaptive transmission period—the time interval between subsequent transmissions is selected dynamically according to the heuristic algorithm, with hysteresis assuming a memory size of *r*.

The main simulation parameters are summarized in Table 1.

The different configurations of intra-platoon messaging have been compared in terms of the total use of resources, being understood as the average number of intra-platoon transmissions per simulation run, as well as in terms of the fraction of emergency braking distance violations observed on average per simulation run (as defined on the left-hand side of the inequality (Equation 8)).

Figure 6 and Figure 7 present the average use of resources (number of transmissions) and the fraction of emergency braking distance violations per simulation run, respectively, vs. different average inter-arrivals of the acceleration disturbance z(t). We have compared the proposed heuristic adaptive approach with the different memory depth used in hysteresis (r={0,200,500,1000} ms), and the conventional approach using fixed messaging interval, with the period ranging from 200 ms to 1 s, representing different resources consumption scenarios. As expected, the best performance in terms of the avoidance of emergency braking activation is achieved with very frequent transmissions, assuming fixed intervals of 200 or 300 ms. However, these scenarios represent a greedy approach on the resources, where over 10,000 transmissions are performed per simulation run. On the other hand, transmission with a fixed low periodicity that is close to 1 s results in frequent emergency braking activation (up to over 15% of simulation time), simply meaning that the controller fails to achieve safe platoon operation. However, when applying an adaptive approach following the proposed heuristic approach, a performance that is similar to frequent fixed interval transmission can be achieved in terms of the avoidance of emergency braking, while using far fewer resources. For the adaptive configuration without hysteresis (r=0ms) hardly any emergency braking is observed, while the use of resources is comparable with fixed interval transmission using a 600 ms period. When applying the hysteresis approach, that which is intended to provide even higher robustness, one can notice the actual fraction of emergency braking activation slightly increases compared with the approach without memory, indicating that a conservative approach, relying on selecting rather more frequent transmissions, results in a lower stability of platoon control. The reason for such an observation may be the fact that a long-term differences in periodicity used for different platoon vehicles may result in the loss of string stability, thus leading to safety degradation. Thus, there is no justification for the use of hysteresis, as it results in higher resources consumption at no improvement (or even degradation) of safety.

The observations from Figure 7 are confirmed with the analysis of the distance error evolution in a selected single simulation run. Figure 8 and Figure 9 show examples of such an evolution for a configuration with a fixed interval of 500 ms, and for the adaptive approach with no memory (no hysteresis), respectively. One can notice in Figure 8 that the fixed interval approach using moderate resources consumption may result in the activation of emergency braking when a series of acceleration changes occurs, leading to an increase in the distance error between the leader vehicle and the first follower over the threshold value of 2 m. In the example run presented in Figure 8, five such occurrences can be observed for the first platoon pair (between the leader and the first follower), corresponding to approximately 3.4% of the total simulation time. The other platoon vehicles manage to keep a much smaller distance error, with the maximum value only slightly exceeding 1 m, as with the fixed interval approach, the string stability of the platoon in ensured, and so for each next vehicles’ pair, the error is becoming smaller. The situation is slightly different when the adaptive approach is used, as shown in Figure 9. Here we observe no distance error threshold violations, with the maximum inter-vehicle distance of approximately 1.9 m being observed for the first pair in the platoon, and so emergency braking is never activated. On the other hand, the distance error for vehicles’ pairs that are located closer to the platoon tail may actually exceed in certain situations the distance error observed for the leader–first follower pair. Such observations can be made in Figure 9 in two time regions when accounting for the positive distance error, around 32 s and 525 s, with the maxima of approximately 1.52 m and 1.42 m being noted for the distance between vehicles 3 and 4. These distance errors result from a sudden change in the leader vehicle, from significant deceleration (approximately 2.5 ${m/s}^{2}$) to small acceleration (up to 0.3 ${m/s}^{2}$). The vehicles closer to the tail of the platoon adjust their accelerations already upon the reception of the leader’s message, with the parameters related to the preceding vehicle being received significantly later (even with a couple of seconds of time difference), which results in a decrease in the distance. This is due to the use of different communication intervals selected with the heuristic approach that may lead to a temporary loss of string stability. However, an important observation is that even such a temporary loss of stability does not result in the violation of the distance error threshold, indicating that the platoon is able to maintain a safe operation all of the time.

Another observation that can be made in Figure 9 is that the adaptive algorithm introduces more significant negative distance error values, which represent a situation of an increased distance between the vehicles. Such behavior can be also considered as a negative effect, as the platooning gains diminish with the increase in inter-vehicle distances, where, with the air drag increasing, the reduction in fuel consumption (that can reach between 7% and 15%, depending on the inter-vehicle distance [3]) will be smaller. Moreover, the reliability of V2V communications is lower at higher ranges. Depending on the wireless communication protocol used, a significant drop in the reception rate of the leader packets can be observed [9,31], which imposes limitations on the maximum platoon length (number of vehicles). Typically, an unacceptable reliability with DSRC can be observed already at a distance of 100 m in the case of heavy traffic, with C-V2X performing slightly better [9]. On the other hand, the main reason for the increase in inter-vehicle distances is that in (Equation 8), we put the constraint only on the minimum distance between the vehicles, with the aim of maintaining safety. Modifying the constraint to also account for the negative error values would mitigate this effect; however, at the cost of increased resources use.

## 7. Discussion

The presented numerical results clearly indicate that it is possible to find a tradeoff between the controller performance and the consumption of wireless communication resources. Depending on the road situation, more or less frequent transmissions may be required. Therefore, the adaptive selection of transmission periods for intra-platoon communications show great potential in reducing the use of wireless resources while maintaining safe platoon operation. It has been shown that the amount of used resources can be reduced by a factor of two, maintaining the same safety level (a similar fraction of emergency braking activation).

The advantage of the proposed adaptive transmission period approach is that it can be easily implemented in the state-of-the-art V2X communication systems. With DSRC, as it operates on an event-based manner, where the arrival of a new packet starts the channel access procedure, the proposed scheme can be implemented in the facilities layer, simply regulating the interval between generation of subsequent packets. Alternatively, it can be implemented in the MAC layer with selective dropping of queued packets according to the chosen messaging interval. When it comes to C-V2X and the scheduling grants mechanisms, each selection of a new messaging interval may result in the need for the negotiation of a new grant. However, in the case where the new interval is a multiple of a previous one, simply omitting th eselected transmission opportunities specified by the existing grant can be used.

The presented work, as only a heuristic approach has been proposed and evaluated, can be still further improved to achieve even higher gains, as a temporary loss of controller stability might be observed in certain situations for the current solution. Such a situation can be considered as dangerous when it comes to safety, as at some point the loss of stability may result in an uncontrollable inter-vehicle gap closing. Moreover, it may result in lower gains in terms of a reduction in fuel consumption due to an increase in air drag when the inter-vehicle distances increase. Furthermore, the unstable behavior of the controller in the vehicles closer to the platoon tail may have a significant impact on the reliability of wireless communications, as with greater distances, the number of successful receptions drops. Therefore, some measures should be taken to avoid such situations and to improve the proposed heuristic. Extending the constraint (Equation 8) to account also for the negative distance error values (i.e., aiming at maintaining platooning efficiency) may increase the stability of the controller; however, at the cost of increased consumption of resources. With the approach imposing two-sided bounds on the distance error one can expect that any change in acceleration, no matter if an increase or decrease is observed, will result in the increased consumption of resources to notify the following vehicles of the change without excessive delays. In a scenario with frequent acceleration changes (even if these are minor), this would diminish the gains of the proposed approach, simply providing similar results to a frequent fixed-interval transmission. In such a case, gains would be observed only if a platoon moves with a constant speed for a longer period of time. On the other hand, one can consider using adaptive inter-vehicle target distance, thus increasing the range of acceptable distance errors (as it depends on the difference between the inter-vehicle target distance and the emergency braking activation distance) and providing more flexibility. Additionally, asymmetric bounds may be considered, with a larger distance error deviation being allowed for the increasing inter-vehicle distances (negative values). In such a case, the expected inter-vehicle target distance, the bounds, and the resource consumption can be jointly adapted, depending on the resources availability and the road situation. However, more detailed studies on such an adaptive approach are left for future work. Furthermore, a “less sensitive” approach to message generation can be also considered, similar to the one proposed for cooperative perception mechanism [27], where the information update is sent only if a significant change in parameters is detected (e.g., a change in acceleration, speed, or position by more than a predefined value).

Another option to maintain stability within the platoon may be a cooperative selection of the communication intervals. In such a case, the transmission intervals could be selected jointly by a central entity (e.g., co-located with the platoon leader), and then distributed to all platoon members. However, this approach would require a knowledge of the parameters of all platoon vehicles in the central entity, as well as a very reliable information distribution mechanism. Such information could be distributed in a similar way to the cooperative perception mechanism, where each platoon member would also include in its message the information on the known other members’ parameters. Additionally, messages could be filtered before being relayed, e.g., based on their age or their relevance in the adaptation process, similarly to the approach proposed in [32].

The proposed approach definitely requires further evaluation in a more sophisticated scenario. The considered system model assumed perfect communications between the platoon vehicles, with only some latency being introduced. In a real platooning scenario, such an assumption is certainly not valid as communication outages may occur, especially when considering the communication links between the leader and the vehicles located at the tail of the platoon. Therefore, the next step in this work should be to extend it with a more realistic communication model, accounting for packet drops and temporary communication outages. Finally, more realistic engine and acceleration models can be considered, accounting for the so-called engine lag and different vehicle acceleration capabilities, depending on its velocity (e.g., using the linear decay model).

## 8. Conclusions

This work investigated the aspects of the efficient use of radio resources in an autonomous platooning employing wireless communications. We considered a quasi-periodic exchange of information, where the interval between subsequent transmissions by a platoon vehicle is selected dynamically in an event-dependent way. An optimization problem focusing on the minimization of the number of wireless intra-platoon transmissions was introduced, with a simultaneous safety-based constraint on the expected number of violations of the minimum inter-vehicle distance applied. We derived a recursive formula representing the changes of distance error between the first pair of vehicles in a platoon over time, aiming at a reformulation of the constraint. As the optimal solution for the stated problem cannot be directly found, we proposed a heuristic algorithm employing the prediction of the inter-vehicle distance error evolution over a finite time window, performing an exhaustive search over the transmission intervals set to select the one securing the longest time until the predicted activation of emergency braking when the minimum distance threshold is violated. The proposed approach and the heuristic solution were then evaluated using numerical simulations, with the results showing the advantages of the adaptive transmission scheduling over fixed-interval communications. Based on the presented results, a conclusion can be drawn where it is possible to reduce the use of radio resources with the proposed adaptive approach while maintaining safe platoon operation. However, a more detailed analysis of platoon behavior is required, particularly focusing on the string stability of the platoon using adaptive transmission intervals. Furthermore, the work can be extended by considering an imperfect communication model and accounting for random drops of data packets, as well as constraints on the resource allocation mechanisms applied in V2X communications.

## Figures and Tables

**Figure 2 sensors-23-00060-f002:**
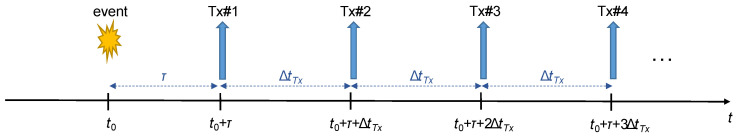
A single platoon vehicle’s transmission pattern, with the initial latency τ of the first message and the update period $\Delta t_{Tx}$ between the transmission of subsequent messages.

**Figure 3 sensors-23-00060-f003:**
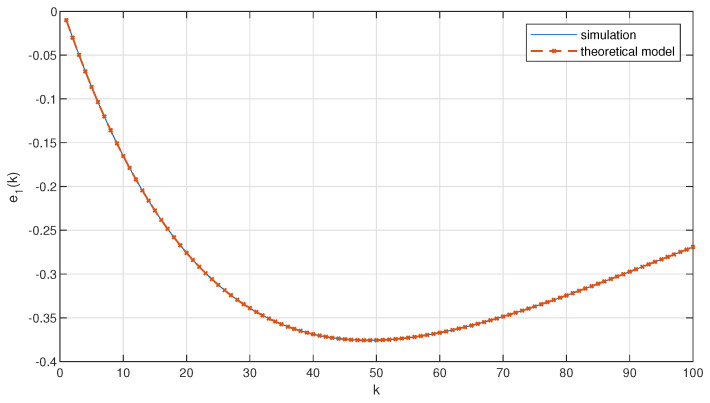
Distance error between leader and the first follower for a single disturbance.

**Figure 4 sensors-23-00060-f004:**
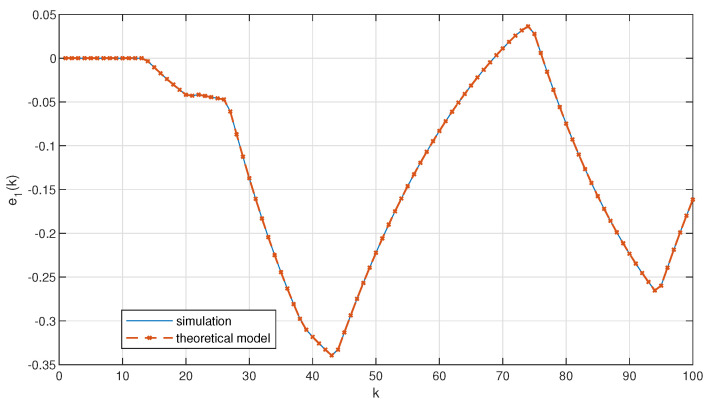
Distance error between leader and the first follower for multiple random disturbances.

**Figure 5 sensors-23-00060-f005:**
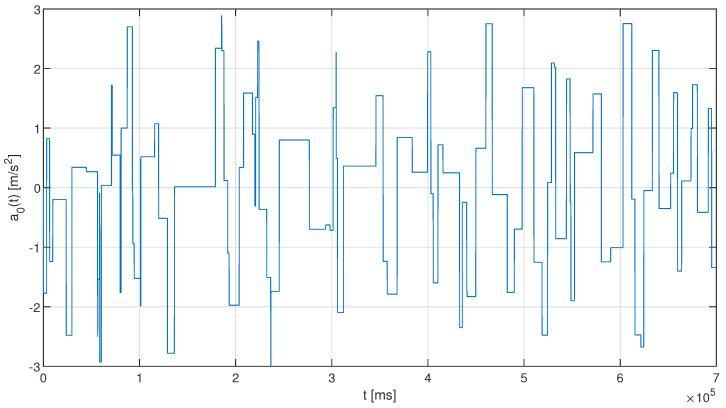
Example of leader acceleration changes (based on $z_{3}$ random variable) in a simulation run.

**Figure 6 sensors-23-00060-f006:**
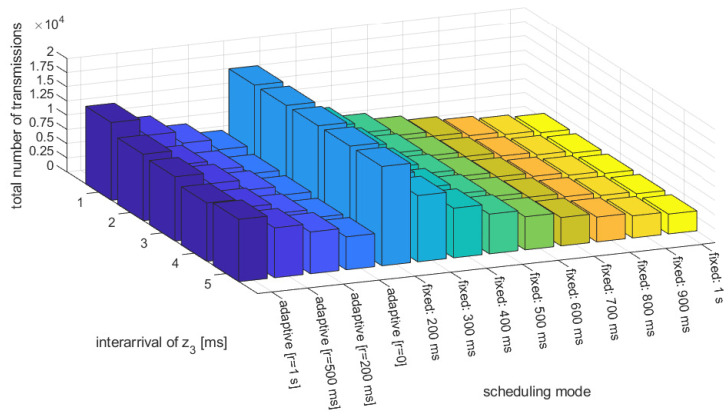
Total number of intra-platoon transmissions per simulation run (700 s) vs. selected transmission period and mean inter-arrival of $z_{0}$ random variable.

**Figure 7 sensors-23-00060-f007:**
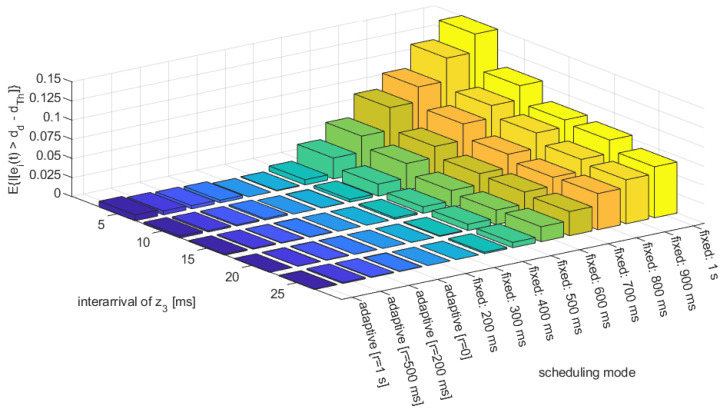
Fraction of emergency braking distance violations per simulation run (700 s) vs. selected transmission period and mean inter-arrival of $z_{0}$ random variable.

**Figure 8 sensors-23-00060-f008:**
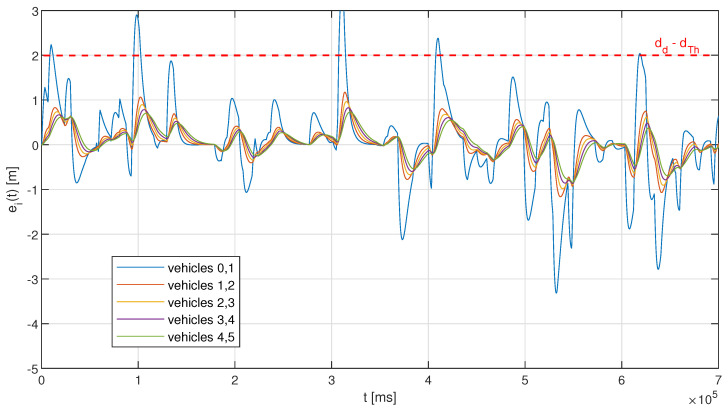
Example of inter-vehicle distance error evolution in a simulation run for fixed periodic transmission with $\Delta t_{Tx}=500\;\mathrm{ms}$; numbers in the legend represent vehicles’ positions in the platoon, with 0 indicating the leader, and 5 denoting the last vehicle (the tail of the platoon).

**Figure 9 sensors-23-00060-f009:**
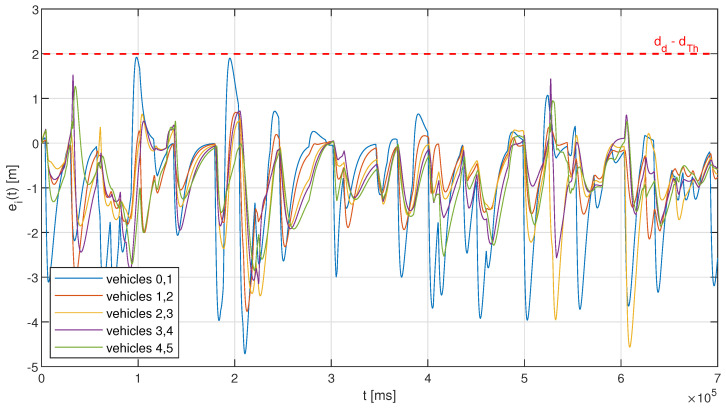
Example of inter-vehicle distance error evolution in a simulation run for adaptive periodic transmission; numbers in the legend represent vehicles’ positions in the platoon, with 0 indicating the leader, and 5 denoting the last vehicle (the tail of the platoon).

**Table 1 sensors-23-00060-t001:** Simulation parameters.

Parameters	Values
Number of platoon vehicles	6
Inter-vehicle spacing in platoon ($d_{d}$)	3 m
Intra-platoon message periodicity ($T_{d}$)	{20 , 50, 100, 200, 500, 1000} ms
Simulation time step	1 ms
Maximum acceleration ($a_{\max}$)	4 m/s${}^{2}$
Minimum acceleration ($a_{\min}$)	−4 m/s${}^{2}$
Maximum velocity	30 m/s
Maximum leader acceleration disturbance ($z_{\max}$)	3 m/s${}^{2}$
Minimum leader acceleration disturbance ($z_{\min}$)	−3 m/s${}^{2}$
Leader acceleration disturbance mean inter-arrival time	{5,10,15,20,25} s
Platoon controller constants ($\left\{\alpha_{1},\alpha_{2},\alpha_{3},\alpha_{4},\alpha_{5}\}\right)$	{−0.04, −0.3, −0.1, 0.5, 0.5}
Heuristic algorithm prediction time span ($t_{\mathrm{Th}}$)	50 s
Heuristic algorithm hysteresis memory size (*r*)	{0, 200, 500, 1000} ms
Emergency braking activation distance ($d_{\mathrm{Th}}$)	1 m
Number of simulation runs per point	50
Single simulation run duration	700 s

## Data Availability

No new data were created or analyzed in this study. Data sharing is not applicable to this article.

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
