# Peer review of "Autonomous Controller-Aware Scheduling of Intra-Platoon V2V Communications"

_sensors, 2022, doi:10.3390/s23010060_

Round 1

Reviewer 1 Report

Overall the motivation seems reasonable and the proposal seems valid.

The paper is fine to follow although some text needs refinement.

My major concern is that the whole proposal focuses on the level of vehicle without looking into a lower level, e.g., message, and thus the gain is limited.

It's better to include some comparison or at least some discussions on the possiblity of combining this work with those focusing on lower levels, include but not limited to:

1. “Analysis of message generation rules for collective perception in connected and automated driving,” in IEEE Intelligent Vehicles Symposium

2. "AICP: Augmented Informative Cooperative Perception", in IEEE TITS

Author Response

We thank the Reviewer for his/her valuable comments, we have addressed them carefully. Please see the attachment.

Author Response

(The authors gave the same response as above.)

Reviewer 3 Report

1. Deepen the section related works with other references.

2. Indicate the reasons why proposed heuristic approach was used in Matlab and not another computational tool.

3. Deepen in the development of the paper the problem on the string stability of the platoon using adaptive transmision intervals. 

4. In section 7 discussion explain in a more quantitative way the results presented in figures 8 and 9 respectively.

Author Response

(The authors gave the same response as above.)

Round 2

Reviewer 1 Report

The authors have revised the draft a lot and the current form is much improved.
